# Optimization of DNA Extraction from Individual Sand Flies for PCR Amplification

**DOI:** 10.3390/mps2020036

**Published:** 2019-05-07

**Authors:** Lorena G. Caligiuri, Adolfo E. Sandoval, Jose C. Miranda, Felipe A. Pessoa, María S. Santini, Oscar D. Salomón, Nagila F. C. Secundino, Christina B. McCarthy

**Affiliations:** 1Centro Regional de Estudios Genómicos, Facultad de Ciencias Exactas, Universidad Nacional de La Plata, La Plata 1900, Argentina; lgcaligiuri@gmail.com; 2Departamento de Informática y Tecnología, Universidad Nacional del Noroeste de la Provincia de Buenos Aires, Pergamino, Buenos Aires 2700, Argentina; 3Laboratorio de Vectores, Secretaría de Calidad de Vida, Municipalidad de Posadas, Posadas, Misiones 3300, Argentina; sandoval@inti.gob.ar; 4Instituto Gonçalo Moniz, Fundação Oswaldo Cruz, Salvador, Bahia 40296-710, Brasil; jmiranda@bahia.fiocruz.br; 5Instituto Leonidas e Maria Deane, Fundação Oswaldo Cruz, Manaus, Amazônia 69057-070, Brasil; favpessoa@gmail.com; 6Centro Nacional de Diagnóstico e Investigación en Endemoepidemias, Administración Nacional de Laboratorios e Institutos de Salud, Ministerio de Salud, Buenos Aires 1063, Argentina; mariasoledadsantini@gmail.com; 7Instituto Nacional de Medicina Tropical, Ministerio de Salud de la Nación, Puerto Iguazú, Misiones 3370, Argentina; odanielsalomon@gmail.com; 8Laboratory of Medical Entomology, René Rachou Research Institute, Fundação Oswaldo Cruz, Minas Gerais 30190-002, Brazil; secundinon@gmail.com

**Keywords:** sand fly, DNA extraction, calcium, PCR, lysis buffer, *Lutzomyia*

## Abstract

Numerous protocols have been published for extracting DNA from phlebotomines. Nevertheless, their small size is generally an issue in terms of yield, efficiency, and purity, for large-scale individual sand fly DNA extractions when using traditional methods. Even though this can be circumvented with commercial kits, these are generally cost-prohibitive for developing countries. We encountered these limitations when analyzing field-collected *Lutzomyia* spp. by polymerase chain reaction (PCR) and, for this reason, we evaluated various modifications on a previously published protocol, the most significant of which was a different lysis buffer that contained Ca^2+^ (buffer TESCa). This ion protects proteinase K against autolysis, increases its thermal stability, and could have a regulatory function for its substrate-binding site. Individual sand fly DNA extraction success was confirmed by amplification reactions using internal control primers that amplify a fragment of the *cacophony* gene. To the best of our knowledge, this is the first time a lysis buffer containing Ca^2+^ has been reported for the extraction of DNA from sand flies.

## 1. Introduction

Various protocols have been published for the extraction of DNA from phlebotomines, including methods that eliminate DNA-associated proteins by using detergents and salts [1,2,3], or with proteinase K and detergents [4], and others that also add extraction steps with phenol-chloroform and precipitation with alcohol [5,6]; commercial DNA extraction kits [7,8]; and the use of Chelex-100 resin [9,10]. Nevertheless, the small size of the sand flies (around 3 mm long) can be an issue, especially in studies that require analysis on an individual basis, such as parasite infection, variability, and population genetics. In particular, these large-scale individual DNA extractions using traditional methods usually yield poor results in terms of efficiency, quantity and purity, which in turn affect PCR success and DNA conservation. This can be circumvented by the use of commercial kits [3], particularly in terms of purity. Notwithstanding, in developing countries, an extensive use of the latter is cost-prohibitive and, consequently, traditional protocols become indispensable.

In our studies we were interested in detecting parasite infection in *Lutzomyia* spp. sand flies by PCR amplification [11]. Nevertheless, as parasite DNA is not necessarily expected, we first needed to confirm that the DNA extraction had been successful, to ensure that negative results were not due to a poor extraction. In our studies this was done with internal control primers that amplify a fragment of the constitutive *Lutzomyia cacophony* gene [12,13]. We previously used a traditional DNA extraction method with pools of 5 and 10 field-captured *L. longipalpis* sand flies. The protocol, which we here refer to as pAC, uses detergent (SDS), proteinase K and phenol-chloroform extraction ([14] and Acardi personal communication). The DNA extracted from these pools of sand flies yielded the expected results consistently when subjected to the internal control PCR. Nevertheless, when we used pAC to process individual sand flies, we found that amplification was poor and inconsistent (i.e., internal control PCR results were variable). For this reason, we decided to evaluate various modifications and, as this method eliminates DNA-associated proteins with proteinase K, we focused on this first crucial step. A number of researchers have reported that calcium ions activate proteinase K and that they are required for the enzymatic action of the protein [15,16,17]. Even though another study disputes the reduction in proteolytic activity of proteinase K when calcium is absent, it concedes that calcium-free proteinase K precipitates irreversibly in the presence of EDTA, leading to a reduced effective concentration [18]. Because of this, we decided to add calcium to the lysis buffer (here referred to as buffer TESCa). DNA extracted from individual sand flies using this and other variations we implemented, produced consistent and successful results in the amplification reactions. To the best of our knowledge, this is the first time a lysis buffer containing Ca^2+^ has been reported for the extraction of DNA from sand flies. 

## 2. Experimental Design

In the following scheme (Scheme 1) we show the main variations that were assayed to optimize DNA extractions from one sand fly (for details see “Section 5”, Appendix B, Appendix C, Appendix D and Appendix E, and Table 1):

## 3. Final Procedure

See also Scheme 2 in Section 5.2.

### 3.1. Lysis and Elimination of Proteins. Time for Completion: ~8 h, 8 min

#### 3.1.1. Homogenization of Sand Fly

Aliquot sufficient volume of buffer TESCa (30 mM Tris-HCl pH 8; 10 mM EDTA; 1% SDS, 5 mM CaCl2; see “Section 4.3”), i.e., 500 μL per sample, and add proteinase K (pK) (to the aliquot) to a final concentration of (0.42 μg/μL).Place one adult sand fly in a 1.5 mL microcentrifuge tube, and add 50 µL of buffer TESCa + pK.
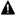
**CRITICAL STEP:** Grind the sand fly thoroughly for 8 min with a Teflon micropestle. To avoid cross-contamination between samples, the micropestle must be cleaned and autoclaved after each grinding (i.e., one micropestle per sample).

#### 3.1.2. Cell Lysis, and Protein Denaturation and Digestion

4.
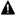
**CRITICAL STEP:** Add 450 µL buffer TESCa + pK (to a final volume of 500 µL), vortex for 1 min, and incubate at 50 °C for 8 h, vortexing for 1 min every 30 min.

### 3.2. Extraction with Solvents. Time for Completion: ~25 min

#### 3.2.1. First Extraction

5.
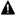
**CRITICAL STEP:** Add 500 µL chloroform:isoamyl alcohol (C:IAA) (24:1 *v*/*v*) and mix vigorously by inversion for 2 min. Immediately centrifuge at 14,000 rpm for 5 min. Transfer the supernatant (~480 µL) to a new 1.5 mL microcentrifuge tube.

#### 3.2.2. Second Extraction

6.
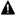
**CRITICAL STEP:** Add 500 µL C:IAA (24:1 *v*/*v*) and mix vigorously by inversion for 2 min. Immediately centrifuge at 14,000 rpm for 5 min. Transfer the supernatant (~460 µL) to a new 1.5 mL microcentrifuge tube.

#### 3.2.3. Third Extraction

7.
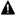
**CRITICAL STEP:** Add 700 µL C:IAA (24:1 *v*/*v*) and mix vigorously by inversion for 2 min. Immediately centrifuge at 14,000 rpm for 5 min. Transfer the supernatant (~400 µL) to a new 1.5 mL microcentrifuge tube.

### 3.3. DNA Precipitation. Time for Completion: ~35 min

#### 3.3.1. Addition of Salt and Alcohol

8.Add 0.1 volumes (~40 µL) 3 M Sodium Acetate (NaOAc) pH 5.2 and 2.5 volumes (~1 mL) 100% ethanol, and gently mix by inversion for 1 min.


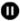
**PAUSE STEP** and **OPTIONAL STEP:** We found that after adding NaOAc and ethanol, results improved when the sample was immediately centrifuged (i.e., was not incubated at all). Nevertheless, due to the length of the previous stages (~9 h), the protocol can be paused here and the sample stored overnight (ON) at −20 °C.

#### 3.3.2. Centrifugation

9.Centrifuge at 14,000 rpm for 20 min and discard the supernatant by inversion.10.Add 500 µL 70% ethanol and centrifuge at 14,000 rpm for 5 min. Discard the supernatant by inversion and dry the pellet at 50 °C for 5 min. Resuspend the pellet in 10 µL double-distilled water.

## 4. Materials, Equipment, and Reagents Setup

### 4.1. Materials

TRIS buffer (NH_2_(CH_2_OH)_3_, 121.14 g/mol) (Anedra, Tigre, Argentina; Cat. no.: AN00915709)Hydrochloric acid (HCl, 36.46 g/mol) (Biopack, Buenos Aires, Argentina; Cat. no.: 9632.08)Sodium Dodecyl Sulfate (SDS, C_12_H_25_NaO_4_S, 288.38 g/mol) (Anedra, Tigre, Argentina; Cat. no.: AN219483180)EDTA (C_10_H_16_N_2_O_8_, 292.24 g/mol) (Anedra, Tigre, Argentina; Cat. no.: AN00605609)Calcium chloride dihydrate (CaCl_2_·2H_2_O, 147 g/mol) (Anedra, Tigre, Argentina; Cat. No.: AN6456)Proteinase K (Fermentas-Thermo Fisher Scientific, Waltham, MA, USA; Cat. No.: #EO0491)Double-distilled water (ddH_2_O)Chloroform (CHCl_3_, 119.38 g/mol) (Cicarelli Laboratorios, San Lorenzo, Argentina; Cat. no.: 1116110)Isoamyl alcohol (Anedra, Tigre, Argentina; Cat. no.: AN00659925)Sodium acetate (CH_3_COONa, 82.03 g/mol) (Anedra, Tigre, Argentina; Cat. No.: AN00651808)Glacial acetic acid (CH_3_COOH, 60,05 g/mol) (Anedra, Tigre, Argentina; Cat. No.: AN6082)Absolute ethanol (C_2_H_6_O, 46.07 g/mol) (Biopack, Buenos Aires, Argentina; Cat. no.: 1654.08)

### 4.2. Equipment

Teflon micropestle (Eppendorf-Fisher Scientific, Leicestershire, UK; Cat. no.: 10683001)Vortex (Denville Scientific, Metuchen, NJ, USA; Cat. no.: Vortexer S7030)Water bath (Jiangsu Jinyi Instrument Technology Company Limited, Shanghai, China; Cat. no.: SHZ-88)High-speed bench-top centrifuge (Heal Force, Shanghai, China; Cat. no.: Neofuge 15)Micropipettes p1000, p200, p20 (Gilson, Middleton, WI, USA; Cat. nos.: F144566, F144565, and F144563)

### 4.3. Reagents Setup

#### Buffer TESCa

Composition: 30 mM Tris-HCl pH 8; 10 mM EDTA; 1% SDS; 5 mM CaCl_2_.

Calculate the necessary volumes for each stock solution. Add and mix the Tris-HCl pH 8, EDTA, and CaCl_2_, autoclave, and then add the SDS.

Below we give an example (Table 2):

## 5. Results

As previously mentioned, we found that internal control PCR results were variable for individual sand flies processed with the pAC protocol (results not shown). For this reason, we assayed various modifications to optimize DNA extractions from one sand fly (Scheme 1). The quality and quantity of the DNA extracts were measured using an AmpliQuant AQ-07 Spectrophotometer, but we found there was no correlation between the amount of DNA quantitated and the success of the PCR reactions. Similarly, a previous study describing the optimization of a DNA extraction procedure from individual human hairs (which poses similar difficulties to those faced by the extraction of DNA from individual sand flies), showed that there was no correlation between the amount of DNA quantitated and the success of STR genotyping, i.e., some extracts were correctly genotyped when quantitation failed to detect any DNA [19]. Hence, and as our main objective was to analyze the DNA extracts in PCR reactions for field studies, success was determined by evaluating each sample in amplification reactions using internal control primers that amplify the IVS6 domain of the *Lutzomyia* constitutive *cacophony* gene (~225 bp) [12,13] (5Llcac and 3Llcac, here referred to as 44F/45R; see Appendix A for detailed PCR conditions). The positive control we used was DNA extracted from a pool of 10 *L. longipalpis* adults from Posadas (Argentina) using the pAC protocol; the negative control was ddH_2_O. The variations we assayed and the effects they produced are mentioned below (see also Scheme 1 and Table 1); for all these extractions we processed field-captured male adult *L. longipalpis* (Posadas, Argentina). 

### 5.1. Assayed Modifications

We first analyzed the effect of minor modifications on the pAC protocol and found that longer incubation times with pK and no incubation at −20 °C (in the DNA precipitation step) in general yielded better results (Appendix B; Table 1). We also found that results improved when mixing by inversion was done vigorously in the extraction with solvents step (Appendix C; Table 1). Nevertheless, as the previous modifications did not determine a consistent improvement, we decided to evaluate changes of greater magnitude (yet including these minor modifications that had produced slight improvements). We decreased the incubation temperature from 58° (original pAC) to 50 °C (our modification) because pK digestion is routinely performed at 50 °C [20], and to move as far away as possible from its inactivation temperature (65 °C) (manufacturer’s recommendation). More importantly, we assayed three different lysis buffers: the original buffer pAC (as control; 10 mM Tris-HCl pH 8; 200 mM NaCl; 5 mM EDTA; 0.2% SDS) (according to Acardi personal communication), another commonly used lysis buffer, here referred to as buffer TES (30 mM Tris-HCl pH 8, 10 mM EDTA and 1% SDS), and this same buffer TES to which we added Ca^2+^ (5 mM), here referred to as buffer TESCa (30 mM Tris-HCl pH 8, 10 mM EDTA, 1% SDS, and 5 mM CaCl_2_; see “Section 4.3”). There were various reasons for evaluating the addition of calcium to the lysis buffer (buffer TESCa). Different studies have reported that calcium ions greatly affect the enzymatic activity of pK [15,16,17] and that enzymatic activity is significantly reduced when they are removed (up to 80%) [15]. Even though another study suggested that calcium ions stabilize the native conformation of the enzyme but do not affect proteolytic activity [18], it showed that Ca^2+^-free pK precipitated irreversibly in the presence of EDTA leading to a much reduced effective concentration [18]. Furthermore, even though Ca^2+^ forms a complex with EDTA in the buffer, it is still capable of interacting with the enzyme [15]. In addition, a previous study found that activation of proteinase K by calcium improved the extraction of DNA from individual human hairs [19]. This same study showed that pK suffered loss of activity when the lysis buffer contained EDTA but no calcium [19]. 

To evaluate these modifications, we processed specimens with the different lysis buffers (pAC, TES, and TESCa) (Table 1) and found that amplification was only positive when the samples were processed with buffers TES and TESCa (Appendix D). Due to these results we decided to further evaluate buffers TES and TESCa and processed more specimens with these buffers. All samples were incubated with pK (0.42 μg/μL) at 50 °C for 8 h, mixing by inversion was done vigorously for the three extractions with C:IAA, and pellets were resuspended in 10 μL ddH_2_O. Due to the length of the first three stages (~9 h), the protocol was paused in the fourth step (i.e., the sample was precipitated ON at −20 °C). Amplification was variable for the samples processed with buffer TES (results were positive for only 2 of the 10 samples; Figure 1), whereas amplification was successful for all the samples treated with buffer TESCa (Figure 1). Chi-square analysis indicated that the association between both treatments (buffers TES and TESCa) and their outcomes was very significant (two-tailed *p* value = 0.0019).

Having determined that buffer TESCa and the previous modifications (incubation with pK at 50 °C, and vigorous mixing during the extraction with solvents), consistently improved DNA extractions, we also analyzed if we could reduce the incubation periods with proteinase K in this new lysis buffer, and eliminate the ON incubation at −20 °C. As we had found previously, longer incubation periods with pK improved PCR amplification, and incubation at −20 °C seemed to have little effect (Appendix E; Table 1). 

### 5.2. Conclusions

To summarize, the main modifications for the final optimized DNA extraction protocol consisted of: (1)an 8-h incubation with proteinase K in buffer TESCa at 50 °C;(2)vigorous mixing by inversion during the extraction with solvents step; and(3)precipitation with alcohol with no ON incubation at −20 °C (Scheme 2). Pellets were resuspended in 10 μL ddH_2_O and a 1:5 dilution was used for the PCR reactions. The complete and detailed optimized protocol is described in “Section 3”.

DNA obtained using our method is suitable for long-term conservation, since individual sand fly DNA extracts were stored at −20 °C and used as a template as much as 6 years later in PCR reactions which yielded positive results.

In conclusion, the above-mentioned changes (the most significant of which was the addition of calcium ions to the lysis buffer) optimized DNA extraction from individual sand flies when compared to the original pAC protocol, and enabled us to consistently obtain positive amplification results with the internal control primers. Moreover, we used this optimized protocol to extract DNA from individual field-captured *Lutzomyia* spp. from Brazil and Argentina, and the internal control amplifications were successful (Appendix F). 

Our results also suggest that pK activity is reduced when the lysis buffer contains EDTA but no calcium (Figure 1), in accordance with a previous study that explored this same solution for optimizing the extraction of DNA from individual human hairs [19], and supporting previous evidence on the dependence of pK activity on the presence of calcium ions [15,16,17]. Furthermore, pK digestion is routinely performed at 50 °C [20], and it has been reported that pK’s activity can increase several fold within the 50 °C to 60 °C range [21]. In this context, buffers pAC, TES and TESCa were tested within pK´s optimal temperature range (50–60 °C), and we only obtained consistently successful results with the lysis buffer that contained calcium (buffer TESCa). Similarly, McNevin et al. [19] used a different lysis buffer (at 56 °C) and also found that DNA extraction only improved when calcium was added to the buffer [19]. This would suggest that, when working within the enzyme´s optimal temperature range, it is the addition of Ca^2+^ to the lysis buffer that improves DNA extraction.

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
