# Peer review of "Optimization of DNA Extraction from Individual Sand Flies for PCR Amplification"

_mps, 2019, doi:10.3390/mps2020036_

Reviewer 1 Report

Dear authors,

I am glad to provide an objective review to the article “Optimisation of DNA extraction from individual sand flies for PCR amplification,” which is targeted to be publish in a journal entitled Methods and protocols (MPs).

The article presents a simple protocol aiming to develop/optimize a sand fly DNA extraction and PCR based diagnostic test to rule out infection by gregarines with no intervention of microscopy detection.

The correct implementation of a simple technique could be extremely useful, since the amounts of DNA that can be extracted from gregarines infecting sand flies is very low and depend on the level of infection.

I do appreciate the development of such a protocol, that I am sure it will be extremely useful for laboratories where research funding is low or for experimental / surveillance field stations where not possibility of other cutting-edge molecular techniques could be used.

Since I am considering this article as a protocol, I am not expecting completely new findings, however the authors shown the optimization of the DNA extraction by the implementation of a lysis buffer with Ca2+ added.

I do not have much comments to make beside the following:

·         The graphical explanation of the protocol is very well addressed and explained.

·         Minor editing of English language and style is needed, for example the word “Optimisation” it is a British spelling and should be consistence with the English style along the text which uses Americanisms.

·         A deep read thought is needed to place comas and grammatical points correctly.

·         Please the “@” for the word “at” everywhere or rephrase the sentences.

·         Figure legends make them shorter and clear. Specially Figure 5, no need the information on lines 275 – 276, make this information clear in the main text.

·         Improve text alignments on tables.

Author Response

Response to Reviewer 1 Comments

I am glad to provide an objective review to the article “Optimisation of DNA extraction from individual sand flies for PCR amplification,” which is targeted to be publish in a journal entitled Methods and protocols (MPs).

The article presents a simple protocol aiming to develop/optimize a sand fly DNA extraction and PCR based diagnostic test to rule out infection by gregarines with no intervention of microscopy detection.

The correct implementation of a simple technique could be extremely useful, since the amounts of DNA that can be extracted from gregarines infecting sand flies is very low and depend on the level of infection.

I do appreciate the development of such a protocol, that I am sure it will be extremely useful for laboratories where research funding is low or for experimental / surveillance field stations where not possibility of other cutting-edge molecular techniques could be used.

Since I am considering this article as a protocol, I am not expecting completely new findings, however the authors shown the optimization of the DNA extraction by the implementation of a lysis buffer with Ca2+ added.

I do not have much comments to make beside the following:

The graphical explanation of the protocol is very well addressed and explained.

Point 1: Minor editing of English language and style is needed, for example the word “Optimisation” it is a British spelling and should be consistence with the English style along the text which uses Americanisms.

Response 1: As suggested by Reviewer 1, the manuscript was revised to ensure consistency with the English style.

Point 2: A deep read thought is needed to place comas and grammatical points correctly.

Response 2: As suggested by Reviewer 1, the manuscript was revised to ensure commas and full stops were placed correctly.

Point 3: Please the “@” for the word “at” everywhere or rephrase the sentences.

Response 3: As suggested by Reviewer 1, “@” was replaced by “at” throughout the text.

Point 4: Figure legends make them shorter and clear. Specially Figure 5, no need the information on lines 275 – 276, make this information clear in the main text.

Response 4: As suggested by Reviewer 1, Figure 5 legend (Figure D1 in the revised manuscript) was shortened.

Point 5: Improve text alignments on tables.

Response 5: As suggested by Reviewer 1, text alignments in the tables was improved.

Reviewer 2 Report

Dear Editor,

In the paper, Caligiuri et al. propose to optimize a cost-effective DNA extraction protocol that can be apply to individual sandflies for large scale studies. They justify this work by the fact that the protocol they routinely use in their lab for pooled sandfly extraction yields variable results (as assessed by PCR amplification of snad fly DNA) on individual sand fly.

The introduction is clear and well written, which gives to think that the paper will provide simple and concise assessment and guideline of their new protocol. However, in my opinion, the rest of the manuscript requires deep restructuring, and reduction of some parts and further validation for some others before being published.

1. Please indicate your experimental design in the experimental design section, what are you going to test and how. A scheme can be useful here (not the scheme of the protocol)

2. Please replace "expected results" by "results". Present your results in subsections like you did. Add titles to the subsections. Group section 2, 3 and 5 in the last subsection of "results", that could be named "optimized protocol".

3. I am not convinced by subsection 4.1 and 4.2. Your claims are based on very few samples and very little visible differences. Please add more samples and/or remove these sections, maybe only keeping a brief mention that you made those choice based on your experience. That would allow to focus on your main finding which is the benefits of adding Ca2+ in the lysis buffer.

4. In section 4.3, I really think that you need to repeat your experience on more samples to claim anything. A simple statistical test like a chi2 could be applied to test the significance of the "buffer" effect on amplification success. Such a test applied on you present results (4 success and 0 failure for TESCA, 2 success and 2 failure for TES) is far from significant (p-value=0.4), which indicate that your result could easlily be explained only by chance. Please increase the number of individuals until you obtain something significant, and then report the result in terms of PCR probability success estimates (with confdence intervals), and p-value.

5. Same thing for section 4.4. I really think that you need to repeat the experiment to say anything. As they are presently shown, i would oly conclude that longer incubation period for lysis seem to slightly improve PCR amplification, and that incubation a -20C seem to have very little effect.

6. PLease end with a short discussion. A few suggestions:

- what is your opinion on the potential health consequences of the use of chloroform ? The latter was banned from many countries. Do you have some alternative in minds ?

- I wonder if the authors have assessed the effect of their protocol on DNA conservation ?

A last remark: I've been using a chelex protocol which is extremly cost and time effective for individual extraction (basically you just put the sand flies in a 96 well plate, add a solution containing Chelex and Pk, let incubate overnight in a thermocycler and then take the surnatant). There is no need for centrifugation, everything can be done with multipipets, and Chelex costs nothing. After dilution to 1:10, I was able to amplify sandfly DNA in almost 100% (with 16S primers) of the case and also use this protocol to successfully amplify sand fly blood meals (with vertebrate specific primers) and Leishmania DNA. Reference: Casquet et al. 2012: Chelex without boiling, a rapid and easy technique to obtain stable amplifiable DNA from small amounts of ethanol-stored spiders (I am not an author of this paper). The only issue with this chlelex protocol is DNA preservation in case you want to store the DNA for long periods.

I also attached a commented pdf with minor suggestions.

Respectfully,

Author Response

Response to Reviewer 2 Comments

 In the paper, Caligiuri et al. propose to optimize a cost-effective DNA extraction protocol that can be apply to individual sandflies for large scale studies. They justify this work by the fact that the protocol they routinely use in their lab for pooled sandfly extraction yields variable results (as assessed by PCR amplification of snad fly DNA) on individual sand fly.

The introduction is clear and well written, which gives to think that the paper will provide simple and concise assessment and guideline of their new protocol. However, in my opinion, the rest of the manuscript requires deep restructuring, and reduction of some parts and further validation for some others before being published.

Point 1: Please indicate your experimental design in the experimental design section, what are you going to test and how. A scheme can be useful here (not the scheme of the protocol)

Response 1: As suggested by Reviewer 2, we have included a new scheme in Section 2 (Experimental Design) which summarises the different variations that were assayed. We think this new figure will simplify interpretation of our results. Nevertheless, since what we tested and how is detailed in Section 4 (Results), Appendices B-D, and Table 1, we did not repeat this information in Section 2.  

Point 2: Please replace "expected results" by "results". Present your results in subsections like you did. Add titles to the subsections. Group section 2, 3 and 5 in the last subsection of "results", that could be named "optimized protocol".

Response 2: When preparing our manuscript, we used the MPs protocol template file as indicated in the “Instructions for Authors”, and did not change the sections and subsections indicated therein, which is why Section 4 is entitled “Expected Results”. Nevertheless, we agree with Reviewer 2 that “Results” is a more appropriate title and will change it accordingly if this is requested by the Editor.

As suggested by Reviewer 2, we have added titles to the subsections in Section 4.

With respect to grouping Sections 2 (Experimental Design), 3 (Procedure) and 5 (Reagents Setup), we agree that it would probably be clearer to do so within Section 4 (Results), but again we mention that the format we followed was that of a Protocol template file provided by MPs. We will group these Sections and place them in the appropriate subsection of Section 4 (Results) upon the Editor’s request.  

Point 3: I am not convinced by subsection 4.1 and 4.2. Your claims are based on very few samples and very little visible differences. Please add more samples and/or remove these sections, maybe only keeping a brief mention that you made those choice based on your experience. That would allow to focus on your main finding which is the benefits of adding Ca2+ in the lysis buffer.

In section 4.3, I really think that you need to repeat your experience on more samples to claim anything. A simple statistical test like a chi2 could be applied to test the significance of the "buffer" effect on amplification success. Such a test applied on you present results (4 success and 0 failure for TESCA, 2 success and 2 failure for TES) is far from significant (p-value=0.4), which indicate that your result could easlily be explained only by chance. Please increase the number of individuals until you obtain something significant, and then report the result in terms of PCR probability success estimates (with confdence intervals), and p-value.

Same thing for section 4.4. I really think that you need to repeat the experiment to say anything. As they are presently shown, i would oly conclude that longer incubation period for lysis seem to slightly improve PCR amplification, and that incubation a -20C seem to have very little effect.

Response 3: When optimising PCR assay conditions, for example to determine the optimal annealing temperature for a certain pair of primers, results are not reported in terms of PCR probability success estimates. That is, a certain temperature gradient will be tested once (maybe twice) on a set of samples, but the results from a single reaction are considered as valid to determine the optimal annealing temperature. Due to the nature of a PCR reaction, it is not required that more assays be performed to obtain statistically significant results. In a similar way, the condition we were assaying here was the template obtained with different protocols. In this sense, results with buffer TESCa were always consistent with the internal control primers (i.e., always positive, as indicated by our analysis of the specimens in the Results Section, and of the 136 individual field-captured Lutzomyia spp. from Brazil and Argentina - see Appendix E -), whereas results with buffers pAC and TES were variable. Moreover, since our objective was to detect parasite infection in sand flies by PCR amplification, with no intervention of microscopy detection, we ruled out those protocols that showed variable results because this implied the extractions were poor and, consequently, we could not ensure that negative results (with the gregarine diagnostic primers) were not due to a poor extraction.

As suggested by Reviewer 2, to allow the reader to focus on our main finding (the benefits of adding Ca2+ to the lysis buffer), Sections 4.1, 4.2 and 4.4 were replaced by general conclusions (p. 10, lines 227-230 and p. 11, lines 285-287), and the detailed assay information contained in those sections was relocated to Appendices B-D.

Point 4: PLease end with a short discussion. A few suggestions:

- what is your opinion on the potential health consequences of the use of chloroform ? The latter was banned from many countries. Do you have some alternative in minds ?

- I wonder if the authors have assessed the effect of their protocol on DNA conservation ?

Response 4: As suggested by Reviewer 2, some of these suggestions were included in the discussion, in particular the effect of our protocol on DNA conservation (p. 12, lines 315-317).  

A last remark: I've been using a chelex protocol which is extremly cost and time effective for individual extraction (basically you just put the sand flies in a 96 well plate, add a solution containing Chelex and Pk, let incubate overnight in a thermocycler and then take the surnatant). There is no need for centrifugation, everything can be done with multipipets, and Chelex costs nothing. After dilution to 1:10, I was able to amplify sandfly DNA in almost 100% (with 16S primers) of the case and also use this protocol to successfully amplify sand fly blood meals (with vertebrate specific primers) and Leishmania DNA. Reference: Casquet et al. 2012: Chelex without boiling, a rapid and easy technique to obtain stable amplifiable DNA from small amounts of ethanol-stored spiders (I am not an author of this paper). The only issue with this chlelex protocol is DNA preservation in case you want to store the DNA for long periods.

We thank Reviewer 2 for this very valuable piece of information.

I also attached a commented pdf with minor suggestions.

We thank Reviewer 2 for taking the time to do this and have included most of these minor suggestions in the revised version of the manuscript.

Round  2

Reviewer 2 Report

Dear Editor,

I think that the authors have put some effort to account for my recommendations and that the article has been improved. Although it is at your discretion to publish this manuscript in it present form or not, I still have to major comments:

- I still find that the organization of the manuscript sections is confusing, but I don't know the precise guidelines of the journal regarding that point. The main problem is that the "experimental protocol" aiming at comparing different extraction protocol, and the "optimized protocol" (which is basically the outcome of this first experiment) are kind of presented jointly already in the "procedure" section. This is mixing methods and results to me. The name of the "expected results" section is also inappropriate in my opinion. An "expected result" is a hypothesis that you formulate prior to your experiment in order to guide the interpretation of your actual results. Here the section is used to provide the actual results of the study.

- the argument of the authors regarding the irrelevance of statistical validation in the case of PCR protocols doesn't stand in my opinion. Would you rely on an experiment in which only 1 sand fly would have been tested for each extraction method ? I don't think so, and me neither, because we know that there is some part of stochasticity in PCR success, and that one cannot rely on one trial only to assess the effect of a treatment. So how many should we choose ? 1, 2, 3, 4 sand flies for each treatment ? This is the precise role of statistical tests to measure in a standardized way whether results significantly deviate from what could happen by chance only. Classical, simple, tests may be viewed as conservative in the case of very well controlled laboratory experiment, but since you cannot really judge yourself whether the conditions are that controlled (and you do have variability in your results, with 2 success and 2 failure in the same buffer treatment group), I still think you should use them. I don't think that clinical trials testing the efficiency of drugs, although being conducted in very controlled conditions, could validate any result without proper statistical testing. Of course, the stakes are not the same here, but this is just to say: proper statistical testing is not complicated to perform (you would probably obtain something significant with 10 sand flies in each treatment and conducting a chi2 test only requires to fill a 2x2 contigency table), and this would actually validate your findings to the eyes of someone like me (and I guess, many researchers).

Respectfully,

Author Response

Response to Reviewer 2 Comments

Comments and Suggestions for Authors

Dear Editor,

I think that the authors have put some effort to account for my recommendations and that the article has been improved. Although it is at your discretion to publish this manuscript in it present form or not, I still have to major comments:

Point 1: - I still find that the organization of the manuscript sections is confusing, but I don't know the precise guidelines of the journal regarding that point. The main problem is that the "experimental protocol" aiming at comparing different extraction protocol, and the "optimized protocol" (which is basically the outcome of this first experiment) are kind of presented jointly already in the "procedure" section. This is mixing methods and results to me. The name of the "expected results" section is also inappropriate in my opinion. An "expected result" is a hypothesis that you formulate prior to your experiment in order to guide the interpretation of your actual results. Here the section is used to provide the actual results of the study.

Response 1: We have made a major re-structuring of the manuscript in order to make a clearer separation between our results and the final optimised protocol. Namely,

1) “4. Expected Results” was renamed as “3. Results” and now includes two subsections “3.1. Assayed Modifications” and “3.2. Conclusions”;

2) “3. Procedure” was renamed as “4. Final Procedure”;

3) “3.2. Figures, Tables and Schemes” is now a separate section: “5. Figures, Tables and Schemes”; and

4) “2.1. Materials” and “2.2. Equipment” were joined with “5. Reagents Setup” in a separate section: “6. Materials, Equipment, and Reagents Setup”.

Point 2: - the argument of the authors regarding the irrelevance of statistical validation in the case of PCR protocols doesn't stand in my opinion. Would you rely on an experiment in which only 1 sand fly would have been tested for each extraction method ? I don't think so, and me neither, because we know that there is some part of stochasticity in PCR success, and that one cannot rely on one trial only to assess the effect of a treatment. So how many should we choose ? 1, 2, 3, 4 sand flies for each treatment ? This is the precise role of statistical tests to measure in a standardized way whether results significantly deviate from what could happen by chance only. Classical, simple, tests may be viewed as conservative in the case of very well controlled laboratory experiment, but since you cannot really judge yourself whether the conditions are that controlled (and you do have variability in your results, with 2 success and 2 failure in the same buffer treatment group), I still think you should use them. I don't think that clinical trials testing the efficiency of drugs, although being conducted in very controlled conditions, could validate any result without proper statistical testing. Of course, the stakes are not the same here, but this is just to say: proper statistical testing is not complicated to perform (you would probably obtain something significant with 10 sand flies in each treatment and conducting a chi2 test only requires to fill a 2x2 contigency table), and this would actually validate your findings to the eyes of someone like me (and I guess, many researchers).

Response 2: We increased the number of samples processed with buffers pAC, TES and TESCa in order to perform a statistical analysis (11 specimens for each treatment; PCR results: TESCa >> 11 positive, 0 negative; TES >> 3 positive, 8 negative; pAC >> 4 positive, 7 negative), and confirmed the significance of the buffer effect on amplification success with a chi2 analysis (TESCa vs. TES: two-tailed P value = 0.0019; TESCa vs. pAC: two-tailed P value = 0.0060). We included a new figure that shows the PCR results for the TES and TESCa samples, as well as the results of the chi2 testing (p. 5, lines 134-143).

This manuscript is a resubmission of an earlier submission. The following is a list of the peer review reports and author responses from that submission.

Round  1

Reviewer 1 Report

This paper describes an optimised protocol for extracting DNA from individual sand flies.  The major finding is that addition of calcium to an extraction buffer with proteinase K enhances DNA yield.  A number of researchers have reported that calcium ions activate proteinase K and that some calcium ion presence is required for the enzymatic action of the protein, eg.:

Betzel et al. (1988) Three-dimensional structure of proteinase K at 0.15 nm resolution, Eur. J. Biochem, 178, 155–171.

Bajorath et al. (1988) The enzymatic activity of proteinase K is controlled by calcium, Eur. J. Biochem. 176, 441–447.

Bajorath et al. (1989) Longerange structural changes in proteinase K triggered by calcium ion removal, Nature 337, 481–484.

Another study disputes the reduction in proteolytic activity of proteinase K without calcium but concedes that calcium-free proteinase K precipitates irreversibly in the presence of EDTA, leading to a reduced effective concentration:

Muller eta al. (1994) Crystal structure of calcium-free proteinase K at 1.5 A resolution, J. Biol. Chem. 269(37), 23108–23111.

Extraction of DNA from individual sand flies poses similar difficulties faced by extraction of DNA from individual human hairs and the solution presented in this paper (activation of proteinase K by calcium) was also explored in:

McNevin et al. (2005) Short tandem repeat (STR) genotyping of keratinised hair Part 2. An optimised genomic DNA extraction procedure reveals donor dependence of STR profiles.  Forensic Science International, 153, 247–259.

The paper would benefit from discussion of this historical background.

The only differences I can see in Figure 1 are visible PCR product at 248 bp in lanes 5, 6, 10, 11,12; faintly visible PCR product in lanes 4, 8; and no visible PCR product in lanes 3, 7, 9.  This suggests to me that, if incubating at -20 oC, this should only occur for 2-4 hours and if incubating at room temperature, this could occur for 3+ hours.  Could the incubation time (in hours) be included in Figure 1, as they are for Figure 5

Figure 5 shows a faintly visible PCR product in lane 9 suggesting that longer incubation times do not always yield more DNA.

Author Response

Response to Reviewer 1 Comments

This paper describes an optimised protocol for extracting DNA from individual sand flies.  The major finding is that addition of calcium to an extraction buffer with proteinase K enhances DNA yield.

Point 1: A number of researchers have reported that calcium ions activate proteinase K and that some calcium ion presence is required for the enzymatic action of the protein, eg.:

Betzel et al. (1988) Three-dimensional structure of proteinase K at 0.15 nm resolution, Eur. J. Biochem, 178, 155–171.

Bajorath et al. (1988) The enzymatic activity of proteinase K is controlled by calcium, Eur. J. Biochem. 176, 441–447.

Bajorath et al. (1989) Longerange structural changes in proteinase K triggered by calcium ion removal, Nature 337, 481–484.

Another study disputes the reduction in proteolytic activity of proteinase K without calcium but concedes that calcium-free proteinase K precipitates irreversibly in the presence of EDTA, leading to a reduced effective concentration:

Muller eta al. (1994) Crystal structure of calcium-free proteinase K at 1.5 A resolution, J. Biol. Chem. 269(37), 23108–23111.

Extraction of DNA from individual sand flies poses similar difficulties faced by extraction of DNA from individual human hairs and the solution presented in this paper (activation of proteinase K by calcium) was also explored in:

McNevin et al. (2005) Short tandem repeat (STR) genotyping of keratinised hair Part 2. An optimised genomic DNA extraction procedure reveals donor dependence of STR profiles.  Forensic Science International, 153, 247–259.

The paper would benefit from discussion of this historical background.

Response 1: This is an excellent suggestion by reviewer 1 and shows very thoughtful insight into the paper. This historical background was mentioned in the Introduction section (p. 2, lines 66-72), and thoroughly addressed in the Expected Results section (p. 9, lines 232-243; p. 11, lines 313-317).

Point 2: The only differences I can see in Figure 1 are visible PCR product at 248 bp in lanes 5, 6, 10, 11,12; faintly visible PCR product in lanes 4, 8; and no visible PCR product in lanes 3, 7, 9.  This suggests to me that, if incubating at -20 oC, this should only occur for 2-4 hours and if incubating at room temperature, this could occur for 3+ hours.  Could the incubation time (in hours) be included in Figure 1, as they are for Figure 5

Response 2: As suggested by Reviewer 1, we have included this observation (p. 7, lines 189-191), as well as the time in hours in Figure 1 (p. 8).

Point 3: Figure 5 shows a faintly visible PCR product in lane 9 suggesting that longer incubation times do not always yield more DNA.

Response 3: As suggested by Reviewer 1, we have included this observation (p. 10, lines 287-288; p. 11, lines 289-290).

Reviewer 2 Report

Dear authors,

Although the interesting subject, I suggest that the content presented in this manuscript (just the final protocol with modifications) be published as part of material and methods of some other study, such as analysis of parasite infection in Lutzomyia spp. as mentioned in introduction. Besides that, there are some major points that should be revised. 

Line 45: the study of parasite infection not necessarily requires analysis on an individual basis. The minimum infection rate can be used  to this kind of study.

Line 164: the information about dilution (1:25) is no relevant. If authors want to mention dilution of DNA it is important to determine the concentration first. It would be interesting to use as positive control, DNA extracted from one individual sand fly using commercial kits, since the objective of the study was to test modifications on pAC protocol. How can I test modifications on a protocol using the own protocol as control?

Line 169: authors say “in the original lysis buffer” – the composition lysis buffer used by Acardi et al., 2010 is different from the buffer used by authors.

Line 174: authors say that “longer incubation times with pK (4 and 8 hours) in general yield better results” – However, in figure 1, we cannot visualize any band in lane 7.

Lines 222, 225, 226: authors test more than one variable in a single experiment. They decrease the incubation temperature from 58ÂşC to 50ÂşC concomitantly with evaluation with three lysis buffers. Were the 3 different lysis buffers tested with incubation at 58ÂşC?

Lines 228, 229, 230: it is a questionable procedure, since the aim of the DNA extraction from sand flies is not the amplification of sand flies DNA, but the amplification of parasites DNA. The dilution can result in a false negative reaction. If authors suppose that undiluted DNA could not be amplified (TES and TESCa) because of inhibitors or impurities, they have to think in other modifications to improve the protocol.

Line 231: there are no content of dilution of DNA in table 1.

Lines 255, 256: according to figure 1, the result obtained with ON incubation at -20ÂşC was not satisfactory since we cannot visualize any result on lane 7. This step could be already excluded.

Reviewer 3 Report

The manuscript entitled "Optimisation of DNA extraction from individual sand flies for PCR amplification” by Caligiuri et al. aimed at highlighting the need for inclusion of calcium chloride (CaCl2) in a lysis buffer for efficient DNA extraction from sand flies. I have the following comments:

Major comments:

 -      The authors ignored the appropriate method of showing the significance of the inclusion of CaCl2in a lysis buffer versus lysis buffer without CaCl2in terms of DNA yield, purity, quality and quantity.  No data showing the quality and quantity of the extracted DNA when different lysis buffer was used. The authors have to show clearly the difference in the DNA quality, purity and quantity using Nanodrop and/or Qubit, and by gel electrophoresis, and not by subjecting the DNA templates to PCR amplification using internal control primers as they did.

-      Most studies have homogenized and lysed individual sand flies in DNA extraction buffer containing150 mM NaCl, 10 mM Tris-HCl, pH 8.0, 10 mM EDTA, 0.1% sodium dodecyl sulfate with 100 g/mL of proteinase K at 37°C, so I do not think changing salt of sodium (NaCl) to salt of calcium (CaCl2) without any justification and clear-cut improvement in the result has any significant contribution to template preparation.

-      The authors sometimes used 1:5 and 1:10 dilutions of the DNA extract for PCR amplification without showing or measuring the DNA concentrations prior to dilution. 

-      It will be good to confirm the amplification consistency of the DNA extract over a short and long period of storage at -20°Cand/or -80°C.

-      Figure 5. I did not see any significant difference in the incubation periods with pK, and overnight or no incubation at -20°C (in the DNA precipitation step). The results are same.

-      Overall, the written English needs a deep review, especially the Experimental Design section 

 Minor comments per heading:

Abstract:

-       Line 29-35: Remove all the citations in the abstract, [1], [2] and Acardi personal communication), [3], [4], and [5,6].

-       Line 31-32:  The statement is not clear and grammatically incorrect “The most significant variation was the use of a different lysis buffer [3] to which added Ca2+(buffer TESCa).

Keywords:

Line 38:  Be consistent. Change Calcium to “calcium”

Introduction:

Line 64-65: This statement is not clear “which was reported as optimized for the extraction of DNA [3] to which added calcium….The statement needs a deep review.

Experimental Design:

What are 5´ and 1´? Please clarify, if its min or hour, be specific. Correct all through the manuscript. 

Line 113: Remove this statement “according to the number of samples you will process”. Instead use “500 Âµl per sample”.

Line 118: Change this statement “(i.e., you should have one micro pestle ready for each sample you are going to process)” to “(i.e. one micro pestle per sample)”.

Line 121:  Remove “reach” (to reach a final volume of 500 µl).

Line 126: Change “during” 2´ to “for” all through the manuscript.

Line 246: Change “submitted” to “subjected”

Line 260: The authors claimed that optimizing DNA extraction protocol for the analysis of parasite infection in Lutzomyiaspp was their main purpose, however, either was the hypotheses tested or any data of successful Leishmaniaamplification from sand flies shown.

Line 261: Please remove “In this sense”

Line 270- 278. All these details are not necessary.

Line 327: The suggestion that Lmigoneiis much smaller than other species analyzed and could account for insufficient DNA for successful amplification, is very wrong and unacceptable, such statement is misleading. The authors should remove the statement. 

Author Response

Response to Reviewer 2 Comments

The manuscript entitled "Optimisation of DNA extraction from individual sand flies for PCR amplification” by Caligiuri et al. aimed at highlighting the need for inclusion of calcium chloride (CaCl2) in a lysis buffer for efficient DNA extraction from sand flies. I have the following comments:

Major comments:

Point 1: The authors ignored the appropriate method of showing the significance of the inclusion of CaCl2in a lysis buffer versus lysis buffer without CaCl2in terms of DNA yield, purity, quality and quantity.  No data showing the quality and quantity of the extracted DNA when different lysis buffer was used. The authors have to show clearly the difference in the DNA quality, purity and quantity using Nanodrop and/or Qubit, and by gel electrophoresis, and not by subjecting the DNA templates to PCR amplification using internal control primers as they did.

Response 1: The main objective for the optimisation we performed was to analyse DNA extracts in PCR reactions for field studies. In this respect, for our research purposes, the success of the procedure was ultimately determined by a positive result when using internal control primers, even if the quality and quantity of the DNA extracts had been previously confirmed. Further, we did measure the quality and quantity of various of the DNA extracts using an AmpliQuant AQ-07 Spectrophotometer, but many times found there was no correlation between the amount of DNA quantitated and the success of the PCR reactions. This same problem was encountered in a previous study that describes the optimisation of a DNA extraction procedure from individual human hairs, which explored the same solution we show in this manuscript (i.e., the activation of proteinase K by calcium) [1].

Nevertheless, we agree with Reviewer 2 that these points needed clarifying, and have addressed them by including a reference the aforementioned study in the Expected Results section (p. 7, lines 173-176).

Point 2: Most studies have homogenized and lysed individual sand flies in DNA extraction buffer containing150 mM NaCl, 10 mM Tris-HCl, pH 8.0, 10 mM EDTA, 0.1% sodium dodecyl sulfate with 100 g/mL of proteinase K at 37°C, so I do not think changing salt of sodium (NaCl) to salt of calcium (CaCl2) without any justification and clear-cut improvement in the result has any significant contribution to template preparation.

Response 2: As Reviewer 1 also pointed out, different studies have reported that calcium ions greatly affect the enzymatic activity of pK [2–4], and that enzymatic activity is significantly reduced when they are removed (up to 80%) [2]. Even though another study suggested that calcium ions stabilise the native conformation of the enzyme but do not affect proteolytic activity [5], it showed that Ca2+-free pK precipitated irreversibly in the presence of EDTA leading to a much reduced effective concentration [5]. Furthermore, even though Ca2+ forms a complex with EDTA in the buffer, it is still capable of interacting with the enzyme [2]. In addition, a previous study found that activation of proteinase K by calcium improved the extraction of DNA from individual human hairs (which poses similar difficulties to those faced by the extraction of DNA from individual sand flies) [1]. This same study showed that pK suffered loss of activity when the lysis buffer contained EDTA but no calcium [1].

We have included a detailed discussion of this historical background (p. 2, lines 66-72; p. 9, lines 232-243; p. 11, lines 313-317), as suggested by Reviewer 1.

Point 3: The authors sometimes used 1:5 and 1:10 dilutions of the DNA extract for PCR amplification without showing or measuring the DNA concentrations prior to dilution.

Response 3: As we already mentioned in Response 1, we did measure the quality and quantity of various of the DNA extracts using an AmpliQuant AQ-07 Spectrophotometer, but many times found there was no correlation between the amount of DNA quantitated and the success of the PCR reactions. For this reason, templates were assayed undiluted and diluted in PCR reactions. The 1:10 dilutions corresponded to DNA extracted with pAC (Figures 1 and 2). We assayed these extractions undiluted and diluted (1:5, 1:10, 1:25 and 1:50), but found that PCRs were only successful when the extracts were undiluted (p. 7, lines 184-186; p.8, lines 217-218). The 1:5 dilutions corresponded to DNA extracted with TES (Figures 3 and 4) and TESCa (Figures 3-5, B1-B3) and, contrary to what we found for pAC, undiluted DNA extracts did not yield positive results in the PCR reactions (p. 9, lines 254-258), probably due to the presence of inhibitors. For this reason, we tried different dilutions (1:5, 1:10, 1:25 and 1:50), found that inhibitors did not affect amplification as from (1:5) dilution and, consequently, used this dilution henceforth.  

Point 4: It will be good to confirm the amplification consistency of the DNA extract over a short and long period of storage at -20°C and/or -80°C.

Response 4: Even though this analysis was not performed systematically, individual sand fly DNA extracts were stored at -20oC and used as template 3 years later in PCR reactions which yielded positive results.

Point 5: Figure 5. I did not see any significant difference in the incubation periods with pK, and overnight or no incubation at -20°C (in the DNA precipitation step). The results are same.

Response 5: We agree with Reviewer 2 that differences are subtle, but neither did we state that the differences were significant in the manuscript. As Reviewer 1 also concedes, “overall… band intensity decreased as incubation time with proteinase K decreased” and “overall band intensity was greater for the samples that were not precipitated at -20oC”. Nevertheless, as Reviewer 1 pointed out, we included the observation that “if incubating at -20oC, this should only occur for 2-4 hours and if incubating at room temperature, this could occur for 3+ hours”.

Point 6: Overall, the written English needs a deep review, especially the Experimental Design section

Response 6: As suggested by Reviewer 2, the manuscript was reviewed by native English speakers.

Minor comments per heading:

Abstract:

Point 7: Line 29-35: Remove all the citations in the abstract, [1], [2] and Acardi personal communication), [3], [4], and [5,6].

Response 7: As suggested by Reviewer 2, all citations were removed from the abstract.

Point 8: Line 31-32: The statement is not clear and grammatically incorrect “The most significant variation was the use of a different lysis buffer [3] to which added Ca2+(buffer TESCa).

Response 8: As suggested by Reviewer 2, the statement was rewritten (p. 1, lines 29-33)

Keywords:

Point 9: Line 38:  Be consistent. Change Calcium to “calcium”

Response 9: As suggested by Reviewer 2, “Calcium” was changed to “calcium” throughout the text.

Introduction:

Point 10: Line 64-65: This statement is not clear “which was reported as optimized for the extraction of DNA [3] to which added calcium….The statement needs a deep review.

Response 10: As suggested by Reviewer 2, the statement was reviewed and rewritten (p. 2, lines 72-74).

Experimental Design:

Point 11: What are 5´ and 1´? Please clarify, if its min or hour, be specific. Correct all through the manuscript.

Response 11: 5´ corresponds to 5 minutes. As suggested by Reviewer 2, this was corrected and specified throughout the text and in Scheme I – Graphical Abstract.

Point 12: Line 113: Remove this statement “according to the number of samples you will process”. Instead use “500 µl per sample”.

Response 12: As suggested by Reviewer 2, the statement was removed and replaced by “500 µl per sample” (p. 5, lines 123-124).

Point 13: Line 118: Change this statement “(i.e., you should have one micro pestle ready for each sample you are going to process)” to “(i.e. one micro pestle per sample)”.

Response 13: As suggested by Reviewer 2, the statement was changed (p. 5, lines 128-129).

Point 14: Line 121:  Remove “reach” (to reach a final volume of 500 µl).

Response 14: As suggested by Reviewer 2, “reach” was removed (p. 5, line 131).

Point 15: Line 126: Change “during” 2´ to “for” all through the manuscript.

Response 15: As suggested by Reviewer 2, “during” was changed to “for” throughout the text.

Point 16: Line 246: Change “submitted” to “subjected”

Response 16: As suggested by Reviewer 2, “submitted” was changed to “subjected” (p. 11, line 295).

Point 17: Line 260: The authors claimed that optimizing DNA extraction protocol for the analysis of parasite infection in Lutzomyia spp was their main purpose, however, either was the hypotheses tested or any data of successful Leishmania amplification from sand flies shown.

Response 17: As Reviewer 2 correctly mentions, the main objective for optimising the DNA extraction protocol was the analysis of gregarine parasite infection in field-collected Lutzomyia spp. In this respect, after the optimisation we were able to perform this field study using the diagnostic primers we had designed [6], and did obtain gregarine amplification in some of the specimens we analysed. Nevertheless, as this was not the main focus of this paper, we did not include these results because we consider this could be misleading. Instead, we rewrote all the sentences where parasite infection had been mentioned, removing all mention of this (p. 1, lines 29-33; p. 11, lines 317-322) except for one sentence in the Introduction section where it was left solely for historical background (p. 2, lines 52-54).

Point 18: Line 261: Please remove “In this sense”

Response 18: As suggested by Reviewer 2, “In this sense” was removed (p. 11, lines 317-322).

Point 19: Line 270- 278. All these details are not necessary.

Response 19: As suggested by Reviewer 2, the details were removed (p. 12, lines 327-332).

Point 20: Line 327: The suggestion that L. migonei is much smaller than other species analyzed and could account for insufficient DNA for successful amplification, is very wrong and unacceptable, such statement is misleading. The authors should remove the statement. 

Response 20: As suggested by Reviewer 2, this sentence was removed (p. 13, lines 387-389).

 Round  2

Reviewer 3 Report

Major comments:

1.    Figure 3, the authors claimed that samples extracted using the buffer pAC with pK at 50°C and resuspended in 10 Âµl ddH2O, and used undiluted for the PCR reactions were able to amplify but when 1:5 dilution of the same samples were used as template no amplifications. However, when TESCa was employed, it was opposite. These results require better explanation, the inability to amplify in the case of 1:5 dilution (pAC) was it due to insufficient DNA template or other factors, and that of undiluted template (TESCa) was it due to impurities/inhibitors/high concentration or other factors? The authors need to clarify or explain this, before adopting their final optimized DNA extraction protocol as stated in Line 275-281.

2.    This study (evaluation of all these modifications) should be between the original lysis buffer pAC (10 mM Tris-HCL pH 8; 200mM NaCl; 5mM EDTA; 0.2% SDS) which they claimed that amplification was poor and inconsistent when used to process individual flies, unlike pooled sample flies and TESCa (30 mM Tris-HCL pH 8; 5mM CaCl2; 10 mM EDTA; 1% SDS) buffers. I do not see the essence of including TES buffer. The authors should repeat the figure 3, 4 and 5 experiments using pAC and TESCa buffers with pK at 37°C for 8 hours.

Minor comments:

1.     Table 1, change “0.5 hs” to “30 min” and all through the manuscript.

2.     Line 169 [0,42 Âµg/µl], should be “[0.42 Âµg/µl]”

Author Response

Response to Reviewer 2 Comments

Major comments:

Point 1: Figure 3, the authors claimed that samples extracted using the buffer pAC with pK at 50°C and resuspended in 10 µl ddH2O, and used undiluted for the PCR reactions were able to amplify but when 1:5 dilution of the same samples were used as template no amplifications. However, when TESCa was employed, it was opposite. These results require better explanation, the inability to amplify in the case of 1:5 dilution (pAC) was it due to insufficient DNA template or other factors, and that of undiluted template (TESCa) was it due to impurities/inhibitors/high concentration or other factors? The authors need to clarify or explain this, before adopting their final optimized DNA extraction protocol as stated in Line 275-281.

Response 1: As suggested by Reviewer 2, this explanation (which we had already made in Response 3 of our first round of responses to Reviewer 2, but had not included in the manuscript) was added to the manuscript (p. 9, line 258, p. 10, lines 259-261).

Point 2: This study (evaluation of all these modifications) should be between the original lysis buffer pAC (10 mM Tris-HCL pH 8; 200mM NaCl; 5mM EDTA; 0.2% SDS) which they claimed that amplification was poor and inconsistent when used to process individual flies, unlike pooled sample flies and TESCa (30 mM Tris-HCL pH 8; 5mM CaCl2; 10 mM EDTA; 1% SDS) buffers. I do not see the essence of including TES buffer. The authors should repeat the figure 3, 4 and 5 experiments using pAC and TESCa buffers with pK at 37°C for 8 hours.

Response 2: As we mention in the “Expected Results” section of the manuscript, “The variations we assayed and the effects they produced, are mentioned… in chronological order”. The reason for this is that we consider that all the information contained here builds up towards our final optimised protocol, and adds to the conclusions. If we removed our results with buffer TES the obvious question is, what would have happened if we had not added Ca2+ to the buffer? Furthermore, the fact that both pAC and TES showed inconsistent results, and that McNevin et al. (2005) used yet another buffer and found DNA extraction only improved when calcium was added to the buffer [1], would suggest that the addition of Ca+2 to different lysis buffers improves DNA extraction, as is implicit in our conclusions (p. 11, lines 316-320).

Proteinase K digestion is routinely performed at 50oC [2] because at this temperature some proteins will unfold making it easier for pK to degrade them. Further, it has been reported that raising the temperature from 37oC to 50-60oC can increase pK’s activity several fold [3]. In view of this historical background, we consider that repeating the experiments using pAC and TESCa buffers with pK at 37°C would not contribute any significant new data to our optimisation. 

Minor comments:

Point 3: Table 1, change “0.5 hs” to “30 min” and all through the manuscript.

Response 3: We made the change from minutes to hours according to Reviewer 1´s suggestion (Point 2), which we were in agreement with as it allowed for more consistency and easier reading.

Point 4: Line 169 [0,42 µg/µl], should be “[0.42 µg/µl]”.

Response 4: As suggested by Reviewer 2, [0,42 µg/µl] was changed to [0.42 µg/µl] (p. 7, line 182).

 References:

[1]      McNevin, D.; Wilson-Wilde, L.; Robertson, J.; Kyd, J.; Lennard, C. Short Tandem Repeat (STR) Genotyping of Keratinised Hair Part 2. An Optimised Genomic DNA Extraction Procedure Reveals Donor Dependence of STR Profiles. Forensic Sci. Int., 2005.

[2]      Farrell, R.E. Resilient Ribonucleases. In RNA Methodologies; 2010.

[3]      EBELING, W.; HENNRICH, N.; KLOCKOW, M.; METZ, H.; ORTH, H.D.; LANG, H. Proteinase K from Tritirachium Album Limber. Eur. J. Biochem., 1974.